# Fourier Ptychographic Microscopy via Alternating Direction Method of Multipliers

**DOI:** 10.3390/cells11091512

**Published:** 2022-04-30

**Authors:** Aiye Wang, Zhuoqun Zhang, Siqi Wang, An Pan, Caiwen Ma, Baoli Yao

**Affiliations:** 1Xi’an Institute of Optics and Precision Mechanics, Chinese Academy of Sciences, Xi’an 710119, China; wangaiye@opt.cn (A.W.); yaobl@opt.ac.cn (B.Y.); 2University of Chinese Academy of Sciences, Beijing 100049, China; 3CAS Key Laboratory of Space Precision Measurement Technology, Xi’an 710119, China; 4Department of Electronic and Electrical Engineering, University of Sheffield, Sheffield S1 3JD, UK; zhang.zq@usheffield.uk; 5Centre Énergie Matériaux Télécommunications, Institut National de la Recherche Scientifique, Varennes, QC J3X 1S2, Canada; amarywang@outlook.com

**Keywords:** Fourier ptychographic microscopy, alternating-direction method of multipliers, phase retrieval, computational imaging, digital pathology, whole slide imaging

## Abstract

Fourier ptychographic microscopy (FPM) has risen as a promising computational imaging technique that breaks the trade-off between high resolution and large field of view (FOV). Its reconstruction is normally formulated as a blind phase retrieval problem, where both the object and probe have to be recovered from phaseless measured data. However, the stability and reconstruction quality may dramatically deteriorate in the presence of noise interference. Herein, we utilized the concept of alternating direction method of multipliers (ADMM) to solve this problem (termed ADMM-FPM) by breaking it into multiple subproblems, each of which may be easier to deal with. We compared its performance against existing algorithms in both simulated and practical FPM platform. It is found that ADMM-FPM method belongs to a global optimization algorithm with a high degree of parallelism and thus results in a more stable and robust phase recovery under noisy conditions. We anticipate that ADMM will rekindle interest in FPM as more modifications and innovations are implemented in the future.

## 1. Introduction

Pathology is a bridge between basic research and clinical applications. Traditional pathology relies on observation through a microscope eyepiece with the naked eye. It is necessary to switch back and forth between an objective lens with different magnification and shift field of view (FOV) to detect pathological features. For a piece of tissue with a thickness of 1 mm~1 cm, 100~1000 sections may be cut by a slicer. Too many slices are required to be observed. Undoubtedly, there exist multiple problems during this process such as easy omission, misjudgment, subjectivity, and low efficiency [1].

Thanks to the development of digital imaging devices, modern digital pathology adopts digital imaging and mechanical scanning to obtain a high resolution (HR) first and then a large FOV so as to achieve full FOV imaging of a single slice, termed whole slide imaging [1]. This kind of instrument is called digital pathology scanner, and it still has four defects to be overcome. First, the imaging quality is not high enough: due to the drift of electronic devices, the stitching boundary is prone to artifacts. For sparse samples, due to a large amount of blank area and lack of adjacent features, they cannot be spliced, and feature-matching errors tend to cause double image. Second, the application scenarios are limited: due to the narrow depth of field (DOF) of a high numerical aperture (NA) objective, it is limited to histopathology or hematology and cannot be applied in cytopathology for a certain thickness. Third, the efficiency is relatively low: among the four-step workflow of translation, autofocusing, imaging, and stitching, the imaging time accounts for a small proportion and the detector is idling for most of the time. Fourth, the instrument is expensive: its operation is largely dependent on the high-precision electric translation platform and high NA objectives.

Fourier ptychographic microscopy (FPM) [2,3,4,5] was invented in 2013 by Zheng and Yang et al. by introducing the concept of ptychography [6,7] into the reciprocal (Fourier) space, which breaks the trade-off between large FOV and HR with a combination of synthetic aperture radar (SAR) [8,9] and optical phase retrieval [10,11]. The objective can only collect light ranging from a certain angle, characterized by the NA. However, parts of the scattering light with a high-angle illumination can also be collected because of light-matter interaction. The sample’s high-frequency information can be modulated into the passband of the objective lens. Instead of conventionally stitching small HR image tiles into a large FOV, FPM uses a low NA objective innately with a large FOV to stitch together low-resolution (LR) images in Fourier space and finally obtain HR images. Compared with conventional digital pathology, no mechanical scanning is required, and the low NA objective has an innate long DOF and working distance, which ideally solves the above-mentioned problems. Currently, FPM has been written into the *Introduction to Fourier Optics* (4th edition) by Goodman [12]. Given its flexible setup without mechanical scanning and interferometric measurement, FPM has improved rapidly, and it not only acts as a tool to obtain both HR and a large FOV but is also regarded as a paradigm to solve a series of trade-off problems, including whole slide imaging [13,14,15,16,17,18], circulating tumor cell (CTC) analysis [19], high-throughput drug screening [20,21], label-free (single shot) high-throughput imaging in situ [22,23,24], retina imaging [25], 3D imaging [26,27,28], wafer detection [29], HR optical field imaging [30], optical cryptosystem [31] and remote sensing [9,32].

Reconstruction algorithms play an essential role in FPM and have progressed rapidly with the help of ever-improving computing power. The original scheme is termed as Gerchberg–Saxton (GS) [6,10], which is proposed in the framework of alternating projection (AP) and similar with the ptychographic iterative engine (PIE) [7] in ptychography. In brief, the estimated solution is alternatively projected to two constraint sets: modulus constraint (captured images) in the spatial domain and support constraint (finite pupil size) in the Fourier domain. The algorithm finally recovers the object’s phase distribution after repeating the above procedures several times. Although reconstruction algorithms based on AP have proved to be effective for phase retrieval, they are not robust enough to produce high-quality reconstruction results under the condition of non-negligible noise. This can reasonably be attributed to the non-convex nature of phase retrieval problems. Therefore, no global convergence guarantees can be achieved [11,33]. More advanced optimization strategies for FPM phase retrieval have afterwards been discussed and implemented involving the difference map [34], Wirtinger flow [35], Gauss–Newton method [36], adaptive step size [37] and convex relaxation [38].

The solutions mentioned above can generally be classified into two categories: global gradient algorithms and sequential gradient algorithms. The former uses the entire collection of diffraction patterns to update the object function at each iteration, while the latter attempts to update the object function using diffraction patterns one by one. Comparatively speaking, the global gradient algorithms are revealed to be superior in algorithmic robustness and the sequential gradient algorithms are more favored and widely used for their fast convergence and high computational efficiency. The FPM phase retrieval problem can be described as an optimization problem of cost function. Accordingly, algorithms can be further classified as first-order or second-order methods. First-order methods only use the first-order derivative of the cost function to update the object function, and second-order methods realize the update based on both the first- and second-order derivative of the cost function. Table 1 gives an intuitive review of the classification of typical algorithms for FPM. While computing the second-order derivative increases the algorithmic complexity, second-order methods show faster convergence rate and better algorithmic robustness. Wirtinger flow is a typical first-order phase retrieval method and can be approximated to a second-order method when the step size and initial guess are elaborately chosen. Given the above, the second-order sequential Gauss–Newton method shows better performance than other algorithms for its relatively high-quality image reconstruction and low time cost and is therefore widely used. Recently, in 2017, Maiden et al. reported a momentum-accelerated PIE (mPIE) algorithm [39] by introducing the idea of momentum in machine learning community into the original PIE algorithm. This intriguing demonstration proved that mPIE is indispensable when the data is extremely difficult to invert and outperforms other PIE algorithms [7,40] in terms of convergence rate and algorithmic robustness.

In this paper, we explored the possibility of utilizing the concept of alternating direction method of multipliers (ADMM) [41,42,43] in FPM, termed ADMM-FPM algorithm. ADMM is a simple but powerful algorithm widely used in applied statistics and machine learning. As a matter of fact, the use of the ADMM method for solving phase retrieval problems is not novel. Some effort has been made to apply ADMM in ptychography and it turns out that ADMM usually outperforms traditional projection algorithms [44]. Such ideas are recently extended to address blind ptychography phase retrieval problems with the unknown probe simultaneously recovered [45]. The problem is formulated as a unique optimization model based on Poisson noise estimation and then solved by a generalized ADMM method which is guaranteed to converge under mild conditions. 

ADMM-FPM is actually proposed based on the research above with the purpose of further solving blind phase retrieval problems in FPM. We reported an optimization model that employed the variant form of ADMM, a typical extension of standard ADMM. Because the FPM problem is a non-convex feasibility problem, there is no theoretical guarantee that ADMM-FPM will converge to the correct solution. However, numerous simulations and experiments were conducted to prove the convergence of the proposed algorithm with a suitable step size. By analysis, it is known that ADMM-FPM approach belongs to a global algorithm with a high degree of parallelism and hence combines the advantages of both sides (global and sequential). Compared with current algorithms, our method presents significantly better reconstruction performance in the presence of intense noise. 

The reminder of the paper is organized as follows: Section 2 formulates the FPM model as a nonlinear minimization problem, followed by the derivation of the proposed ADMM-FPM algorithm. The results of simulations and experiments conducted to verify the proposed algorithm are presented in Section 3 and Section 4, respectively. Section 5 ends the paper with conclusions and a discussion.

## 2. Principles

### 2.1. Problem Formulation

The demonstration of FPM procedures can be referred to [2,3,4,5] and will not be detailed here. In this paper, we denote the LR intensity image captured at each illumination angle as:(1)$$I_{j}={\left|{ℱ}^{H}\left\{diag\left(p\right)Q_{j}ℱu\right\}\right|}^{2}\in R_{+}^{m},j=1,2,\ldots k$$
where ℱ is the 2D Fourier transform and the superscript *H* denotes a Hermitian conjugate. The object to be constructed is denoted as *u*∈C*^n^.* The operation *diag*(*x*)*y* represents the element-by-element multiplication between two vectors, *x* and *y*. Matrix **Q***_j_*∈R*^m×n^* denotes the downsampling of the object in the Fourier domain corresponding to the *j*-th illumination angle. The pupil function *p*∈C*^m^* can be considered as a circular aperture that imposes constraint because of its finite size. To recover the HR object function from a stack of LR intensity images, we describe the reconstruction process as the following optimization problem:(2)$$\underset{s}{\min}\sum_{j=1}^{N}f_{j}\left(s\right):={\sum_{j=1}^{N}\left\|\left|{ℱ}^{H}\left\{diag\left(p\right)Q_{j}s\right\}\right|-\sqrt{I_{j}}\right\|}_{2}^{2}$$

Here, *s* = *u*∈C*^n^* denotes the object in the Fourier domain. Under regular circumstances, as the lens aberration is almost not completely known, the problem can be further transformed to:(3)$$\underset{p,s}{\min}\sum_{j=1}^{N}f_{j}\left(p,s\right):={\sum_{j=1}^{N}\left\|\left|{ℱ}^{H}\left\{diag\left(p\right)Q_{j}s\right\}\right|-\sqrt{I_{j}}\right\|}_{2}^{2}$$

The corresponding diagram to describe this mathematical and physical process is shown in Figure 1. As the sample sections we observe under microscope are exceedingly thin and can always be approximated to weak-phase objects, it may be tough to effectively separate the pupil function from the object while solving the optimization problem above. In order to suppress the crosstalk between the two, a Tikhonov regularization is considered to be set:(4)$$g\left(s\right)=\gamma {\left\|s-\delta \right\|}_{2}^{2}$$
where *γ* > 0 and *δ* represents the Fourier transform of a flat light field without any phase information. The supplement of a regularization term with priori information helps to stabilize optimization during subsequent operations [46]. The final description of the regularized optimization problem is given as:(5)$$\underset{p,s}{\min}\sum_{j=1}^{N}f_{j}\left(p,s\right)+\frac{1}{2}g\left(s\right):={\sum_{j=1}^{N}\left\|\left|{ℱ}^{H}\left\{diag\left(p\right)Q_{j}s\right\}\right|-\sqrt{I_{j}}\right\|}_{2}^{2}+\frac{\gamma }{2}{\left\|s-\delta \right\|}_{2}^{2}$$

### 2.2. ADMM Solution

The ADMM method is rooted in the 1950s and was originally proposed in the mid-1970s [47,48]. The performance of the ADMM method and its variants [49,50,51,52,53,54] has been studied intensively since its demonstration. It was found that ADMM is well suited to distributed convex optimization problems. Since many important-to-solve problems in statistics and machine learning can be posed in the framework of convex optimization, such as the collection and storage of large-scale modern datasets, the ADMM method gains widespread popularity in the two fields. It is noteworthy that the method takes the form of a ‘decomposition-coordination’ procedure [41]. By introducing new variables, the original optimization problem is decomposed into several subproblems linked by augmented Lagrangian, the local solutions to these subproblems are then coordinated to reach a global solution to the original optimization problem.

Here we show how ADMM can be used to solve the optimization problem for FPM. We introduce the intermediate variables *q_j_* = *diag*(*p*)**Q***_j_ s* and solve:(6)$$\underset{s,\left\{q_{j}\right\}}{\min}\sum_{j=1}^{N}f_{j}\left(q_{j}\right)+\frac{1}{2}g\left(s\right):={\sum_{j=1}^{N}\left\|\left|{ℱ}^{H}q_{j}\right|-\sqrt{I_{j}}\right\|}_{2}^{2}+\frac{\gamma }{2}{\left\|s-\delta \right\|}_{2}^{2}$$
which is equivalent to (5). The augmented Lagrangian function of (6) is defined as:(7)$$ℒ\left(p,q_{j},s,\lambda_{j}\right)=\sum_{j=1}^{N}\left(f_{j}\left(q_{j}\right)+\lambda_{j}\left[diag\left(p\right)Q_{j}s-q_{j}\right]+\frac{\alpha }{2}{\left\|diag\left(p\right)Q_{j}s-q_{j}\right\|}^{2}\right)+\frac{\gamma }{2}{\left\|s-\delta \right\|}_{2}^{2}$$
where *α* > 0 is a penalty parameter and *λ_j_* is the Lagrange multiplier corresponding to the intermediate variables *q_j_*. If we denote *x^k^* as the *k*-th estimated value of *x*, the update of variables is given according to the following order:(8)$$q_{j}^{k+1}=\arg\underset{q_{j}}{\min}ℒ\left(p^{k},q_{j},s^{k},\lambda_{j}^{k}\right)$$
(9)$$s^{k+1}=\arg\underset{s}{\min}ℒ(p^{k},q_{j}^{k+1},s,\lambda_{j}^{k})$$
(10)$$p^{k+1}=\arg\underset{p}{\min}ℒ\left(p,q_{j}^{k+1},s^{k+1},\lambda_{j}^{k}\right)$$
(11)$$\lambda_{j}^{k+1}=\lambda_{j}^{k}+\alpha \eta \left(q_{j}^{k+1}-diag\left(p\right)Q_{j}s^{k+1}\right)$$
where *η* > 0 is the step size appropriately chosen, which determines how fast the local solutions get close to the global solution. We substitute *ω_j_* = *λ_j_*_/_*α* and the scaled-form ADMM iterations can be further expressed as:(12)$$q_{j}^{k+1}=\arg\underset{q}{\min}f_{j}\left(q\right)+\frac{\alpha }{2}{\left\|q-diag\left(p\right)Q_{j}s^{k}+\omega_{j}^{k}\right\|}_{2}^{2}$$
(13)$$s^{k+1}=\frac{\gamma \delta /\alpha +\sum_{j=1}^{N}Q_{j}^{H}diag\left(\bar{p}\right)\left(q_{j}^{k+1}+\omega_{j}^{k}\right)}{\gamma /\alpha +\sum_{j=1}^{N}Q_{j}^{H}{\left|p\right|}^{2}Q_{j}}$$
(14)$$p^{k+1}=\arg\underset{p}{\min}{\left\|q_{j}^{k+1}-diag\left(p\right)Q_{j}s^{k+1}+\omega_{j}^{k}\right\|}_{2}^{2}+\beta {\left\|p\right\|}_{2}^{2}$$
(15)$$\omega_{j}^{k+1}=\omega_{j}^{k}+\eta \left(q_{j}^{k+1}-diag\left(p\right)Q_{j}s^{k+1}\right)$$
where *β* > 0 is the regularization parameter and determines how fast the pupil function should be updated. The closed-form iterations of subproblem (12) and (14) are directly given as follows, detailed derivation of the approximation and Equation (13) is shown in Appendix A.
(16)$$q_{j}^{k+1}=q+diag\left(\frac{1}{1+\alpha }\right){\left[ℱdiag\left(\frac{\sqrt{I_{j}}}{\left|{ℱ}^{H}q\right|}\right){ℱ}^{H}q-{ℱ}^{H}q\right]}_{q=diag\left(p\right)Q_{j}s^{k}-\omega_{j}^{k}}$$
(17)$$p^{k+1}=\frac{\sum_{j=1}^{N}Q_{j}^{H}{\bar{s}}^{k+1}\left(q_{j}^{k+1}+\omega_{j}^{k}\right)}{\sum_{j=1}^{N}Q_{j}^{H}{\left|s^{k}\right|}^{2}Q_{j}+\beta }$$

The termination of the procedure is determined by primal residual and dual residual, whose definitions are given respectively as follows:(18)$$R_{p}^{k}={\sum_{j=1}^{N}\left\|q_{j}^{k}-diag\left(p\right)Q_{j}s^{k}\right\|}_{2}^{2}$$
(19)$$R_{d}^{k}={\sum_{j=1}^{N}\left\|q_{j}^{k}-q^{k-1}\right\|}_{2}^{2}$$

To our knowledge, the process of finding a global solution based on the integration of local solutions should be consistent with the optimization model established in Section 2.1; thus, we consequently rewrite the equivalent form of (18) as follows to simplify the calculation.
(20)$$R_{p}^{k}={\sum_{j=1}^{N}\left\|{ℱ}^{H}q_{j}^{k}-\sqrt{I_{j}}\right\|}_{2}^{2}$$

It is suggested that the reasonable stopping criterion for the method is that the primal and dual residuals must converge to zero according to the optimality conditions. In our simulations, we properly relax the conditions, terminating the iterations when the corresponding normalized error metrics of both residuals converge to a stable level:(21)$$\left(E_{p}^{k}-E_{p}^{k-1}\right)/E_{p}^{k}<\varepsilon^{tol}$$
(22)$$\left(E_{d}^{k}-E_{d}^{k-1}\right)/E_{d}^{k}<\varepsilon^{tol}$$
where *ε^tol^* > 0 is the stopping tolerance and the normalized error metrics are defined as:(23)$$E_{p}^{k}=\frac{{\sum_{j=1}^{N}\left\|{ℱ}^{H}q_{j}^{k}-\sqrt{I_{j}}\right\|}_{2}^{2}}{{\sum_{j=1}^{N}\left\|\sqrt{I_{j}}\right\|}_{2}^{2}}$$
(24)$$E_{d}^{k}=\frac{\alpha \eta {\sum_{j=1}^{N}\left\|q_{j}^{k}-q^{k-1}\right\|}_{2}^{2}}{{\sum_{j=1}^{N}\left\|\lambda_{j}^{k}\right\|}_{2}^{2}}$$

Table 2 offers the suggested ranges for parameters used in the procedure and additional discussion is placed in Appendix A. The pseudo-code listed in Algorithm 1 shows the flowchart of ADMM solution for FPM phase retrieval.

We need to clarify that the method reported in this paper is based on a variant form of standard ADMM. They adopt varying intermediate variables but share similar concepts and procedure. The performances with the standard are not very well according our testing. It should be emphasized that the deduction of ADMM solution above is strictly applicable to convex optimization only. However, the ptychographic phase retrieval problem is still non-convex due to the presence of the Fourier magnitude constraint. Hence, the convergence of ADMM-FPM is not guaranteed, at least in theory. Despite this uncertainty, the proposed method seems still effective in practical FPM implementation when a sufficient amount of overlap among different areas of illumination is provided. In addition, self-adaptive step-size strategy for the ADMM method has been reported with the purpose of providing superior convergence [55,56]. Although the modified method shows great potential for large-scale datasets processing, it fails to generate desired results in FPM phase retrieval. A detailed analysis with simulation comparison is shown in Appendix A.
**Algorithm 1: ADMM Solution for FPM (ADMM-FPM)****Input:***Q_j_*, *I_j_***Output:***s*, *p***Initialize***s*^0^, *p*^0^, *ω*^0^*_j_*, *q*^0^*_j_***for***k* = 1: *Niter* (iterations)  **for**
*j* = 1: *N* (different incident angles)   update *q^k^_j_* according to Equation (16)   update *s^k^* according to Equation (13)   update *p^k^* according to Equation (17)   update *ω^k^_j_* according to Equation (15)   **break when** (21) and (22) are satisfied  **end****end**


## 3. Simulations

To examine the performance of our proposed ADMM-FPM method, we run the scheme on simulated FPM data. According to the suggested parameter ranges given in Table 2, the values of parameters used in this paper are set as: *α* = 0.5, *β* = 1000, *γ* = 0.3, *η* = 1. The simulated FPM platform contains an image sensor with pixel size of 6.5 µm and a 4× objective with NA of 0.1. A programmable 15 × 15 LED array is placed 86 mm above the sample, which provides an incident illumination wavelength of 632 nm and the distance between adjacent LEDs is 4 mm. The number of LED has little impact on the reconstruction quality but is closely related to the spatial resolution; thus, no extra simulations under different numbers of LED should be designed. The raw LR diffraction patterns captured each are limited to a small region of 128 × 128 pixels and the recovered HR images come with a size of 384 × 384 pixels. Under regular circumstances, in order to accelerate convergence, we choose the sampling interpolation of the LR intensity image captured by positive incident illumination as the initial intensity guess [2]. In our simulation, full-one matrix guess proves to yield better reconstruction results for the ADMM-FPM method. NA order is used as the updating sequence of sub-pupils in the Fourier domain, which starts at the center of the Fourier space and then gradually expands outwards [57]. Besides the visual results, we also utilize a quantitative criterion to evaluate the reconstruction performance of the method. The structural similarity (SSIM) [58] measures the extent to which the spatial structural information is close between two images, which is widely used in digital image processing and analysis. The SSIM index is given by:(25)$$SSIM\left(x,y\right)=\frac{\left(2\mu_{x}\mu_{y}+C_{1}\right)\left(2\sigma_{xy}+C_{2}\right)}{\left(\mu_{x}^{2}+\mu_{y}^{2}+C_{1}\right)\left(\sigma_{x}^{2}+\sigma_{y}^{2}+C_{2}\right)}$$
where *x* and *y* are two virtual nonnegative images, which have been aligned with each other, *µ_x_* and *σ_x_* denote the mean intensity and standard deviation (the square root of variance), respectively, of the image vector *x*, and *σ_xy_* represents the correlation coefficient between two image vectors, *x* and *y*. Constants *C*_1_ and *C*_2_ are included to avoid instability when these statistics are very close to zero. The SSIM index ranges from 0 to 1 and higher value indicates that two images are of more similar structural information.

### 3.1. Performance Comparison under Noiseless Conditions

First, we compare the proposed ADMM-FPM method with two widely used phase retrieval algorithms: Gauss–Newton method and mPIE. Gauss–Newton can be classified as a sequential gradient approach, where we set the initial step size *α*^0^ = 1, *β*^0^ = 1. It is noteworthy that the values of step size are not fixed but vary for each iteration. We adopt the concept of self-adaptive step size and update the step size according to the following rule [37]:(26)$$\alpha^{k}=\{\begin{array}{cc} \alpha^{k-1}/2 & \left(E_{}^{k}-E_{}^{k-1}\right)/E_{}^{k}>\varepsilon \\ \alpha^{k-1} & otherwise \end{array}$$

It is the same with the other step size *β*, where *E^k^* denotes the normalized error metric at the *k*-th iteration and *ε* should be a constant much less than 1. It is found that fixed *ε* = 0.01 usually works well to produce desired reconstruction results [37]. The introduction of this strategy allows the algorithm to approach a solution promptly at the early iterations and then gradually converge to a stable level as the step size decreases. Hence, the Gauss–Newton method we use as contrast is actually a modification of the original algorithm intended to improve robustness. Empirical evidence in terms of ptychography reveals that appropriately delaying the update time of reliable variables helps to achieve more stable reconstruction in mPIE algorithm. In our simulations, we are confident about the initial pupil, which is designed to update at the 15th illumination position while the object function is updated at the very beginning. Simultaneously, the momentum parameter for pupil update should accordingly be tuned smaller. Referring to [39], the parameters of mPIE algorithm are properly chosen as: *α* = 0.1, *β* = 0.8, *γ* = 1, *η_obj_* = 0.9, *η_pupil_* = 0.3, *T_pupil_* = 15.

All algorithms are designated to run for 100 iterations. Figure 2(a1,a2) plot the amplitude and phase reconstruction performance of the three algorithms as a function of iteration numbers with their corresponding visual results compared with ground truth demonstrated below. The SSIM values are marked precisely. We find that all of the algorithms achieve convergence and realize stable reconstruction when there is no noise, mPIE algorithm stands out for its fast convergence and high-quality reconstruction results. Our proposed ADMM method can be much slower to achieve convergence with a slightly smaller value of SSIM. However, it is often the case that converging to modest accuracy within a few tens of iterations is sufficient to produce acceptable results of practical use; at least, we can hardly distinguish their differences with the naked eye. The Gauss–Newton method with self-adaptive step size performs roughly in between, merely at a disadvantage in amplitude reconstruction performance. In addition, the reconstruction process involves constant updates to pupil function, which effectively eliminates artifacts derived from objective aberration. Figure 2(b3–e3) intuitively show the introduced pupil for simulation and the reconstructed results of pupil via three algorithms. It can be seen that all algorithms have the ability to correct the aberrations so that a quantitative phase image can be obtained.

### 3.2. Performance Comparison under Noisy Conditions

Ideally, all algorithms based on the forward model demonstrated in Section 2.1 should provide good reconstructions. However, LR images are inevitably corrupted by different degrees of noise in practical implementation of FPM. Hence, it is necessary to analyze the performance of our proposed ADMM method under noisy conditions. Since the brightfield images concentrate the majority of signal power, darkfield images from high-angle illuminations are more susceptible to noise. In order to simulate the actual experimental conditions, we add noise to images whose mean value of intensity is below 0.2. Here, we mainly discuss the cases of Gaussian noise and Poisson noise.

We compare the noise-tolerance capacity of three algorithms under different proportions of Gaussian noise. Figure 3(a1,a2) show the comparison curves of amplitude and phase reconstruction performance. The general trend of the curves is moving downward as Gaussian noise becomes stronger, among which the ADMM method declines at the lowest rate and therefore generates the optimal outcome especially when the noise level is more than 50%. The performance of mPIE algorithm in the presence of Gaussian noise is a considerable departure from its significant advantage presented in the previous simulation. Actually, it can be explained by the fact that the iteration curve of mPIE algorithm becomes extremely unstable with dramatic fluctuations and fails to achieve convergence. The SSIM values are recorded at the iteration where the normalized error metric of the captured intensity images arrives at a plateau despite its irregular oscillation during the later iterations. To intuitively explain the advantage of the ADMM method in noise tolerance, we show one typical group of visual results under 70% Gaussian noise in Figure 3b. The reconstruction results of mPIE algorithm are covered with intense background noise, while the other two algorithms produce a much cleaner result. From the reconstruction results of amplitude, we find that the Gauss–Newton method suffers from slightly more obvious crosstalk between amplitude and phase information than the ADMM method. As is shown in the magnified area [Figure 3(b1–b4)], the ADMM method presents a significant smoothing action, resulting in efficient performance of noise removal.

Poisson distribution is a discrete probability distribution used to describe the statistics of the incoming photons at each pixel. It is assumed that the intensity is proportional to the photon count so that the distribution of intensity can be regarded as Poisson distribution. According to the property of Poisson distribution, a large intensity value measured at a certain pixel indicates severe noise corruption at that pixel. In our simulation, we define a Poisson noise parameter *σ* and generate Poisson noise at each pixel whose mean value equals the ratio of pixel intensity to *σ*, hence *σ* and noise level are negatively correlated.

Since our test proves that noise level *σ* > 0.01 is pretty close to noiseless condition, we only show the comparison curves of three algorithms under Poisson noise ranging from *σ* = 10^−5^ to *σ* = 0.01 in Figure 4(a1,a2). Here, the horizontal axis of coordinate system takes the form of logarithm in order to display the trend of curves more reasonably. Similar behavior can be observed in the graphs as the case of Gaussian noise that the curves extend downward with Poisson noise increasing. The curve of mPIE algorithm is always located at the bottom, implying its equally poor performance under Poisson noise. From somewhere between *σ* = 10^−4^ and *σ* = 10^−3^, the Gauss–Newton method starts to outperform the ADMM method especially in phase reconstruction, which benefits from the self-adaptive step size strategy adopted in Gauss–Newton. At this range, the iteration curve of the Gauss–Newton method goes through a stair-type increase when the step size updates for each iteration [Figure 4(b1,b2)]. Generally speaking, we can safely conclude that our proposed ADMM method holds an advantage when *σ <* 10^−4^. Figure 4d gives one typical group of visual results when Poisson noise is valued *σ* = 10^−4^. For the mPIE algorithm, high-frequency details seem to be oversmoothed and many small-scale features are so blurred that they cannot be distinguished easily. The reconstruction results are also corrupted with strong artifacts, which leads to the uneven distribution of background. The Gauss–Newton method and ADMM method perform similarly from the perspective of visual perception, both with crosstalk between amplitude and phase information as well as artifact corruption, but the ADMM method works better in terms of numerical results. Overall, the removal of Poisson noise seems not so efficient as in Gaussian noise simulations, which we guess might stem from different characteristics of the two kinds of noise. To our knowledge, Poisson distribution can be approximated to Gaussian distribution when the arrival rate of photons is high enough, thus the performance of reconstruction methods under high-intensity Poisson noise should generally resemble that under Gaussian noise. Actually, Gaussian noise and Poisson noise respectively correspond to readout noise and photon shot noise in practical imaging systems. In our simulations, the former is “added to” the darkfield images, while the latter is “generated” based on the pixel intensity according to certain statistical property. Since the simulated Poisson noise added by the above-mentioned method does not certainly conform to strict Poisson contribution, the relationship of approximation cannot be established. Consequently, the performance of the ADMM method is quite different or can be even worse. 

### 3.3. Comparison and Analysis of Algorithm Efficiency

In the discussions above, we run each algorithm for a fixed number of 100 to analyze their trends of reconstruction performance. However, the SSIM values are recorded when the termination conditions are satisfied, hence practical iteration numbers should be fewer than 100 normally. Table 3 gives the iteration numbers and total runtime of three algorithms so that we can study the convergence speed and computational complexity for each algorithm. Typical choices of noise level for the cases of Gaussian noise and Poisson noise generally follow the previous simulations.

When there is no noise, the data is consistent with the convergence rate of each algorithm, as we demonstrate in Section 3.1. Among the three algorithms, the mPIE algorithm spends the shortest time reaching convergence and realizing stable reconstruction. Based on the concept of momentum, mPIE estimates an advance velocity in the right direction by contrasting variables before and after update and boldly takes another step in that direction at the end of each update. This unique idea proves to be an efficient way to accelerate convergence greatly. When noise interferes, the “bold step” might induce the risk that the curve of normalized error metric suddenly oscillates after converging to stabilization (Figure 5). Consequently, the algorithm fails to reach convergence and the reconstruction quality cannot be guaranteed [Figure 4(b1,b2)]. For this case, we terminate the algorithm right as the normalized error metric becomes stable. The iteration numbers and runtime are recorded simultaneously. We should recognize that the normalized error metric is vital to the judgement of algorithm convergence. However, lower error values do not always indicate higher reconstruction quality as the reconstruction results are likely to consist more with the contaminated data and the results are more likely to fall into local limits. As is shown in Figure 5, the error curves of ADMM method do not achieve the lowest stable level, while producing the best noise tolerance performance. Moreover, it should be emphasized that the normalized error metric mentioned here specially refers to normalized “prime error” metric for ADMM method, as given in Equation (23). The calculation of normalized error metric for the other two algorithms also follows the formula. 

The time cost of our proposed ADMM method is the highest whether there is noise or not, which might be attributed to its global and complex framework. On the one hand, the iteration process of the ADMM method consists of alternating update to four variables, while Gauss–Newton and mPIE only update two variables (object function and pupil function) for each iteration. On the other hand, the termination conditions of the ADMM method impose more strict constraint because two factors (primal residual and dual residual) jointly determine the termination of algorithm. By contrast, Gauss–Newton and mPIE terminate as long as a single factor (step size for Gauss–Newton and error metric for mPIE) meets predetermined conditions. The Gauss–Newton method performs at an intermediate level in algorithm efficiency. It should be emphasized that the application of self-adaptive step size strategy introduces no extra computational cost as the error metric is calculated after each cycle of sub-iterations no longer to judge convergence but to determine the step size for the next iteration instead [37]. Given the above, the ADMM method provides the best reconstruction results at the expense of larger iteration number and longer runtime when there is serious noise interference. However, if compared to those global algorithms with hundreds of seconds [35,36], the efficiency of the ADMM-FPM approach is acceptable.

## 4. Experiments

The experimental setup of the real FPM platform is slightly different from our simulations. The programmable 32 × 32 LED array (Adafruit, controlled by an Arduino) is placed 68 mm above the sample, whereas only the central 9 × 9 LEDs are lighted up sequentially for data acquisition. All the data are captured by an 8-bit CCD camera (DMK23G445, Imaging Source Inc., Bremen, Germany, 3.75 μm pixel pitch, 1280 × 960 pixels), and the rest of the parameters chosen for algorithms generally follow the simulations. In practical implementation, the noise level is largely determined by the performance of sensors. In order to emulate experiments with lower-cost sensors, we artificially increase the gain of our camera in the process of data collection, resulting in the captured intensity images with higher brightness and more noise points exposed. Accordingly, the exposure time should properly be adjusted to accommodate the high intensity of brightfield images in case of sensor saturation.

We first image a USAF resolution target for each algorithm. Since we mainly focus on the noise-tolerance capacity of algorithms rather than the resolution improvement, our observation view of interest should not necessarily be limited to the smallest group of features. Here, we extract a small region of 310 × 310 pixels from the full FOV image for analysis, which contains all elements of group 6~9. Figure 6 offers the comparison results of three algorithms under the condition of 20 dB and 30 dB, where the phase reconstruction images display in pseudo-color so that detailed information can be identified clearly. It can be seen that the Gauss–Newton method and mPIE algorithm are not effective enough to remove noise. The background of the Gauss–Newton method is covered with dense noise points evenly, while the noise presents a nonuniform distribution for the mPIE algorithm. Similar phenomena can be significantly observed inside the rectangle bars from the phase reconstruction images. Comparatively, the ADMM method provides a much neater background and internal structure, confirming our conclusions in the simulations under noisy conditions. Additionally, the amplitude reconstruction results of the other two algorithms seem darker than that of the ADMM method. This actually reflects that the contrast of images is not high enough, to some extent indicating that the two algorithms provide poor performance of noise removal. Figure 6(c1,c2) plot the grayscale value of pixels along a certain horizontal section for amplitude reconstruction images under 20 dB and 30 dB gain. Since the contaminated pixels have significant difference in grayscale values compared with the rest of the pixels in the image background, the ADMM method whose curve maintains minimum degree of fluctuation proves to be the most effective to remove noise. Here, the number of pixels does not equal the size of extracted religion as the reconstructed images will be magnified several times. By horizontal contrast, the results of each algorithm prove the correspondence between gain and noise level, namely a larger gain indicates a higher level of noise. At last, we should add that there is no obvious difference in the reconstruction resolution, as we can observe all elements in group 8 through the three algorithms.

Finally, ADMM method is tested on a biological sample (a pathological section of onion scale leaf epidermal cells), where the phase property often contains valuable information for study. Figure 7a provides the full FOV image of the sample under a 4 × /0.1 NA objective with 20 dB gain. The small region of interest extracted from the full FOV image has a size of 210 × 210 pixels (Figure 7(a1)). Figure 7b,c give the comparative reconstruction results under 20 dB and 30 dB gain, respectively. Here, we can determine, according to the original LR image, that the darker reconstruction results of the Gauss–Newton method and mPIE algorithm do not arise from the low contrast of the original data, but from their vulnerability to the interference of noise as in the USAF experiment. We do not adjust the grayscale to a consistent visual level in order to ensure the authenticity of reconstruction results to the greatest extent. All the results are fairly compared within the same scale of grayscale. It can be found that the majority of basic details in the sample have been preserved via the three reconstruction methods, while much noise accumulates around the cell walls of the sample for the Gauss–Newton method and mPIE algorithm. The ADMM method effectively eliminates the noise, and once again outperforms other algorithms with high-quality reconstruction and low-intensity noise distribution.

## 5. Conclusions and Discussions

To summarize, we reported an ADMM-FPM algorithm. Although the proposed method performs quite commonly under noiseless conditions, its prominent potential is experimentally demonstrated in noise tolerance, which trades moderate time cost for the improvement of reconstruction quality. In this paper, we add a regularization term to stabilize optimization for weak-phase objects. Diverse regularizations can be selectively adopted according to the property of objects as our proposed method displays good compatibility with them. How to determine the category that ADMM method belongs to is worthy of our consideration. The grace of ADMM method originates from its unique algorithmic mechanism that the global optimization problem is divided into several subproblems, the solutions of which are updated alternately. The update of object function and pupil is obtained based on the batch update of other variables, normally in the form of accumulation. Unlike conventional global update algorithms, this update is placed in the middle of each iteration period instead of at the end. Compared with sequential update algorithms, the iteration procedure of ADMM method is highly parallel so that it can operate efficiently on GPU or distributed CPUs. In this sense, we might roughly classify the ADMM method as a special global gradient method with the advantages of sequential algorithms.

More modifications and innovations remain to be implemented in this work. We have confirmed that the ADMM method falls short in algorithm efficiency to a certain degree. In terms of noise tolerance performance, we find that ADMM methods are superior only when the noise reaches a certain level. Particularly for the case of Gaussian noise, noise removal usually comes along with blurred detailed information. Hence, under the same computational efficiency, we hope for a more robust strategy to achieve a faster rate of convergence without the sacrifice of reconstruction quality. Ref. [44] has proved the feasibility of a step size update scheme in ptychography when the limit point of the ADMM iteration sequence satisfies certain conditions. This idea might provide some inspiration, though a widely used self-adaptive step-size strategy has been found to not produce the desired results in Appendix A. We emphasize that for any particular problem, it is likely that some variation on ADMM will substantially improve performance. The breakthrough of the ADMM method might be found out in itself by introducing different intermediate variables to create an extended ADMM method. In addition, the performance of the ADMM method suffers from a large set of parameters currently. We are working to give the optimal choices of parameters that make sense mathematically. The demo code is released on our website for noncommercial use and further understanding of this article [59].

## Figures and Tables

**Figure 1 cells-11-01512-f001:**
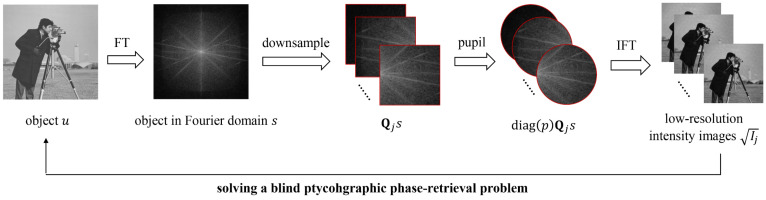
Diagram of FPM reconstruction process. The Fourier transform of the object is firstly downsampled by the illumination matrix, then confined by the pupil function and finally inverse Fourier transformed back to form LR intensity images in the spatial domain. The object is to be reconstructed by solving a blind phase retrieval problem.

**Figure 2 cells-11-01512-f002:**
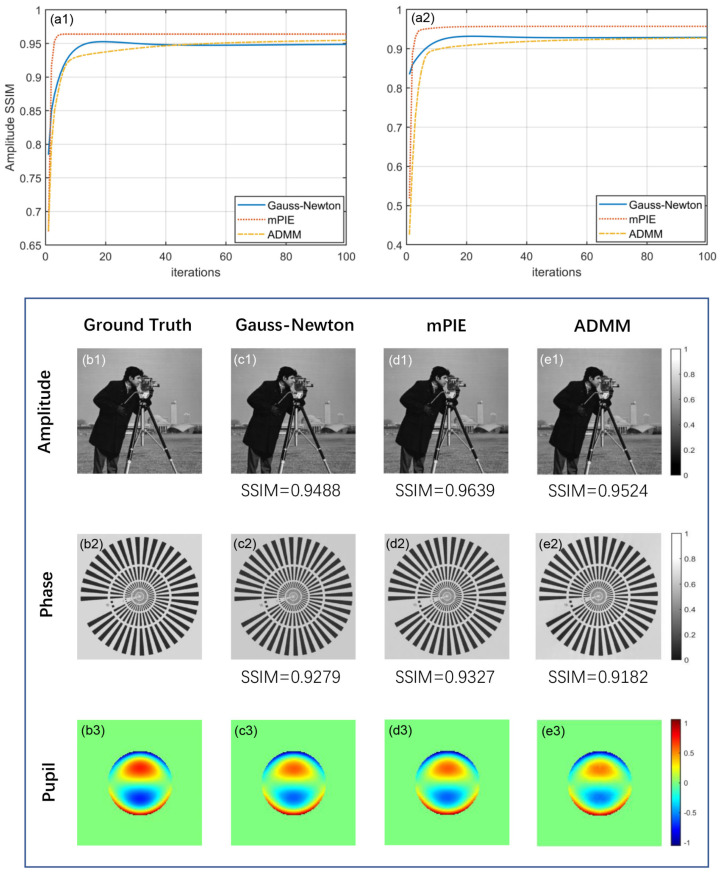
Comparison of simulated reconstruction results under noiseless conditions. (**a1**,**a2**) Amplitude and phase SSIM curves for 100 iterations. (**b1**–**b3**) Simulated ground truth of amplitude, phase and pupil for FPM reconstruction. Corresponding visual reconstruction results: (**c1**–**c3**) Gauss–Newton method. (**d1**–**d3**) mPIE algorithm. (**e1**–**e3**) ADMM method.

**Figure 3 cells-11-01512-f003:**
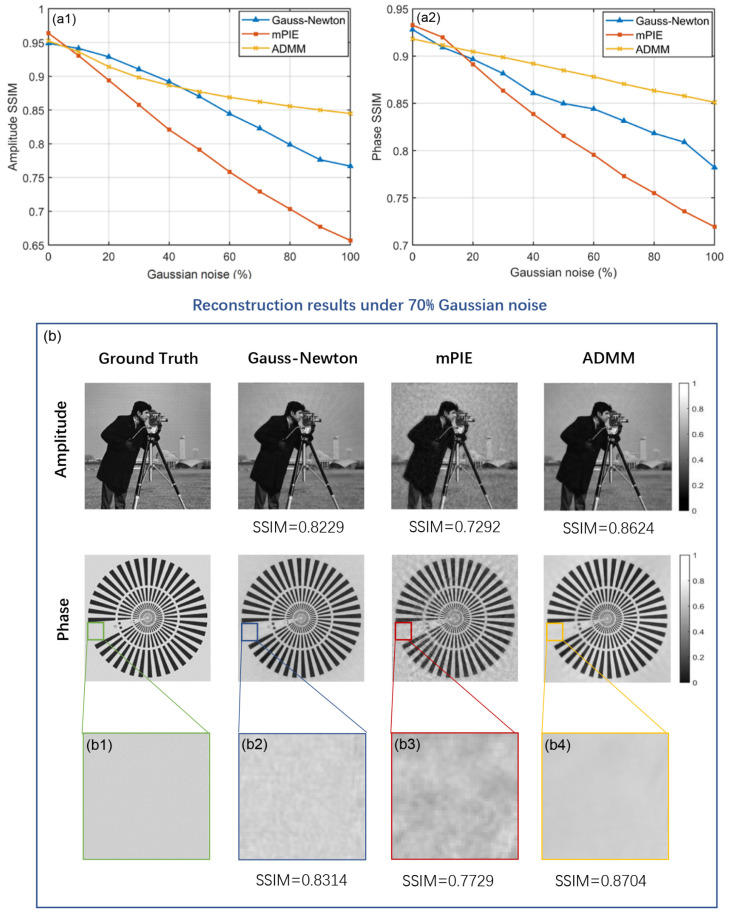
Comparison of simulated reconstruction results under Gaussian noise. (**a1**,**a2**) Amplitude and phase SSIM curves as a function of Gaussian noise level. (**b**) Visual reconstruction results of three algorithms under 70% Gaussian noise with SSIM values marked below the images. Magnified images for the religion of interest in phase reconstruction results are shown in (**b1**–**b4**).

**Figure 4 cells-11-01512-f004:**
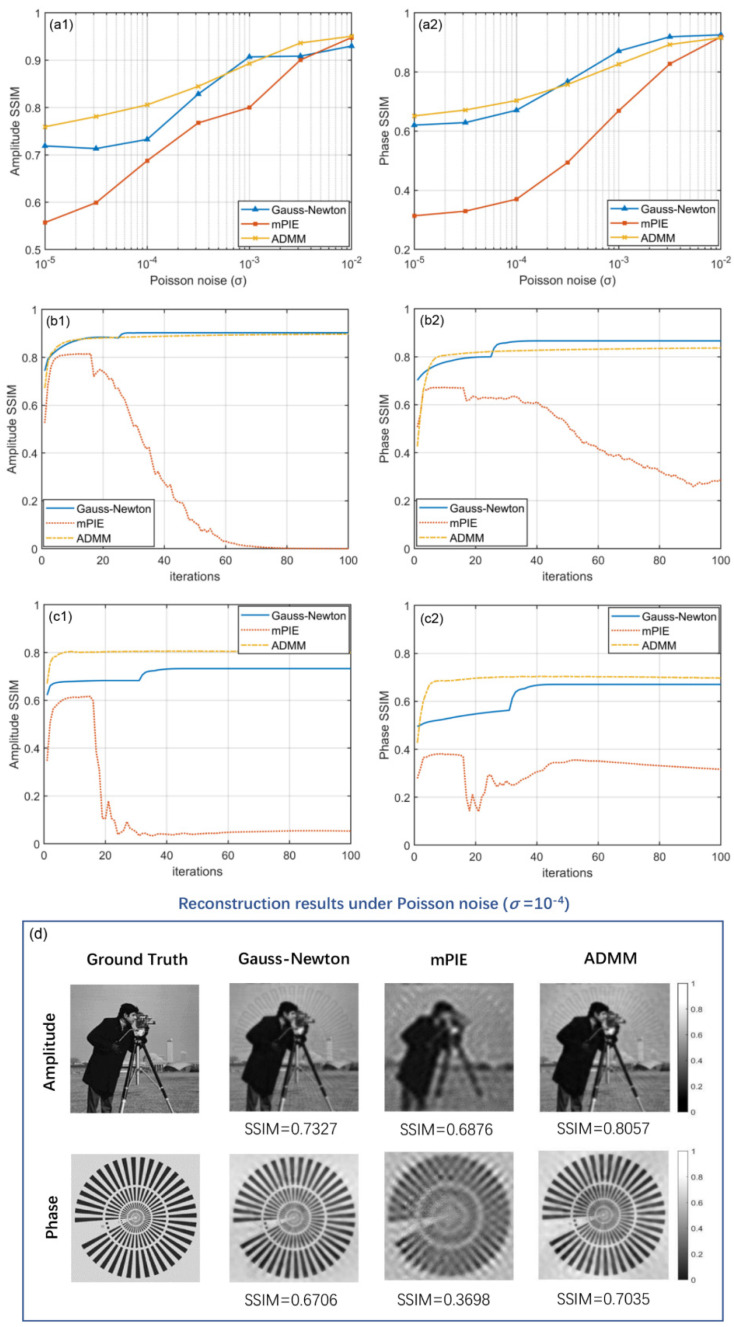
Comparison of simulated reconstruction results under Poisson noise. (**a1**,**a2**) Amplitude and phase SSIM curves as a function of Poisson noise level. (**b1**,**b2**) Amplitude and phase SSIM curves as a function of iteration numbers under Poisson noise (σ = 10^−3^). (**c1**,**c2**) Amplitude and phase SSIM curves as a function of iteration numbers under Poisson noise (σ = 10^−4^). (**d**) Visual reconstruction results of three algorithms under Poisson noise (σ = 10^−4^) with SSIM values marked below the images.

**Figure 5 cells-11-01512-f005:**
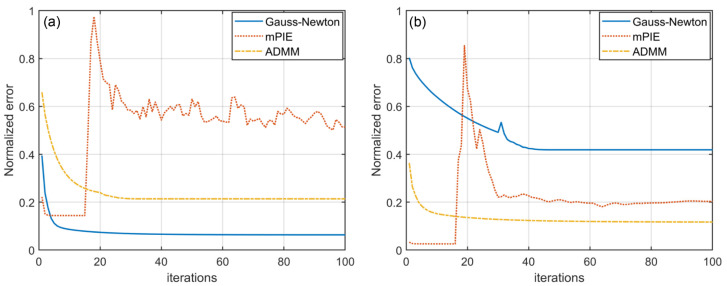
The curves of normalized error metric for three algorithms under (**a**) 50% Gaussian noise and (**b**) Poisson noise (σ = 10^−4^). All algorithms are designed to run for 100 times of iteration.

**Figure 6 cells-11-01512-f006:**
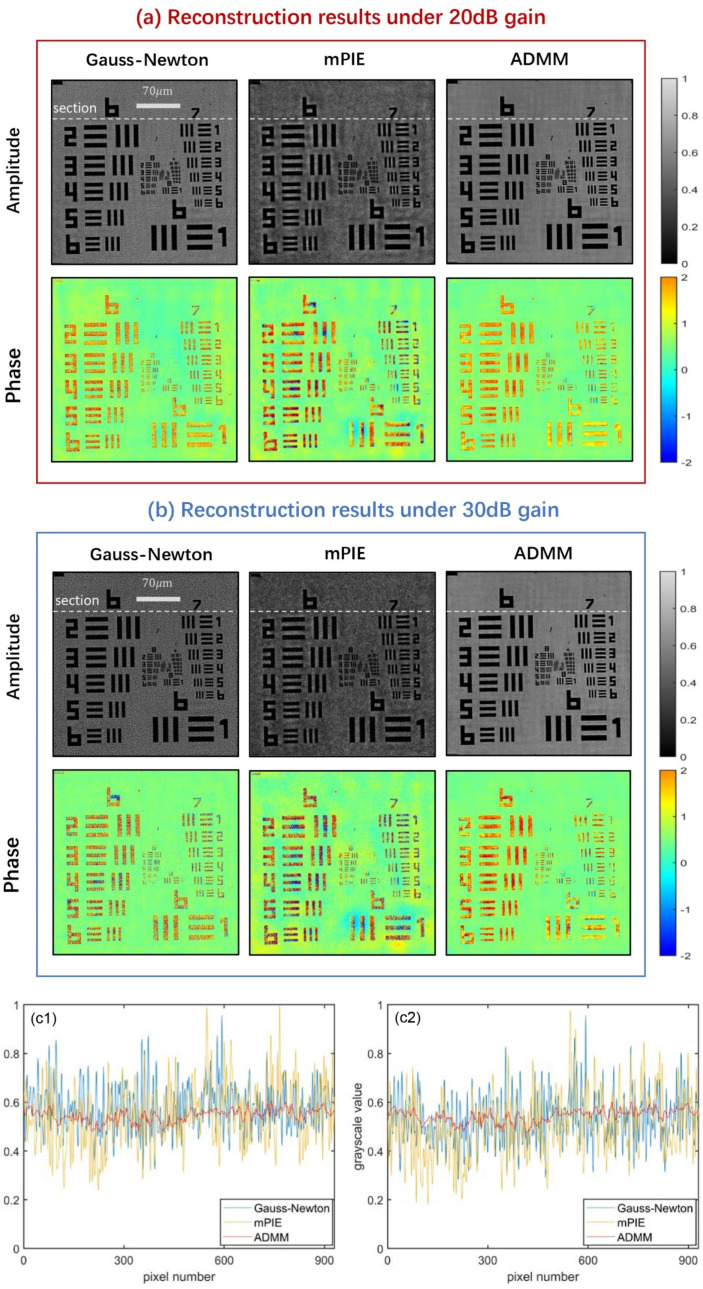
Comparison of experimental reconstruction results for a USAF resolution target under (**a**) 20 dB gain and (**b**) 30 dB gain. (**c1**,**c2**) Plot of the grayscale value of pixels along a certain horizontal section (marked in (**a**,**b**)) for amplitude reconstruction images under 20 dB and 30 dB gain, respectively.

**Figure 7 cells-11-01512-f007:**
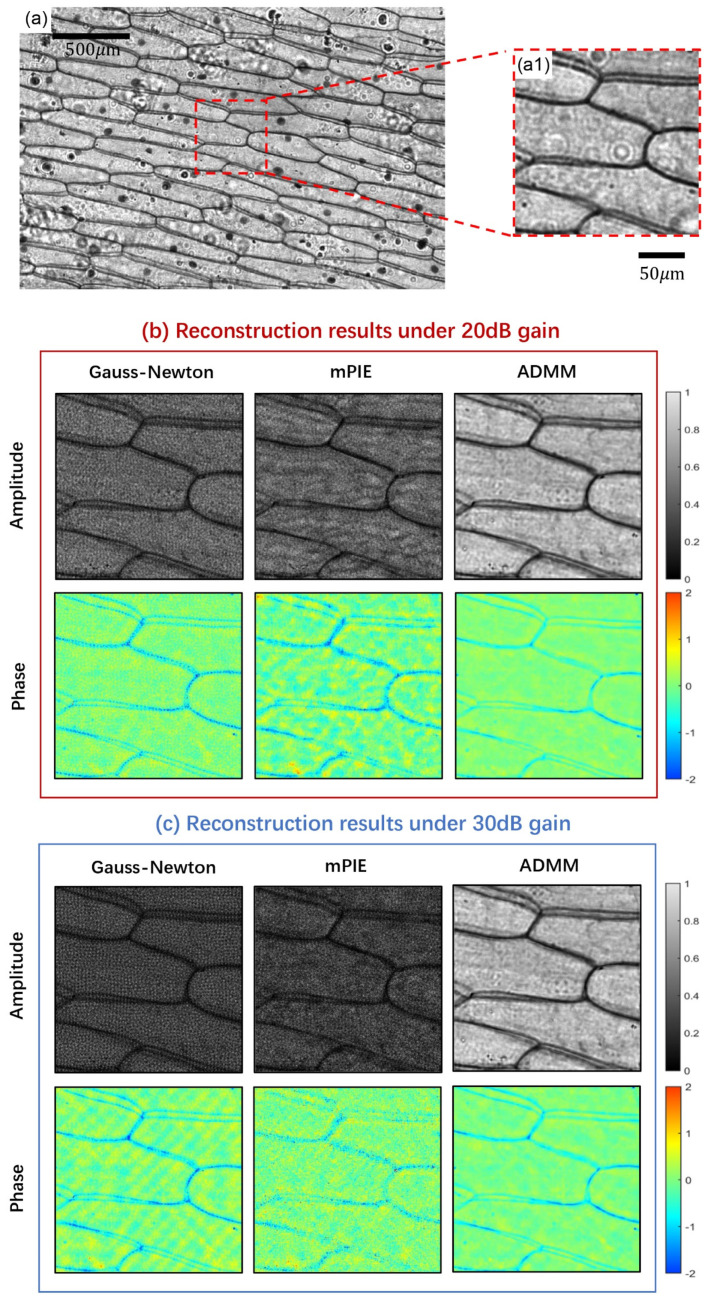
Comparison results of experimental reconstruction for a biological sample (a pathological section of onion scale leaf epidermal cells). (**a**,**a1**) The full FOV image under 20 dB gain and magnified image for the region of interest. (**b**) Reconstruction results under 20 dB gain. (**c**) Reconstruction results under 30 dB gain.

**Table 1 cells-11-01512-t001:** Classification of typical reconstruction algorithms for FPM (Adapted from Ref. [36]).

	First-Order	Second-Order
**Sequential**	G-S algorithm $O^{\left(i,l+1\right)}=O^{\left(i,l\right)}-\frac{1}{{\left|P\right|}_{\max}^{2}}\nabla_{O}f_{A,l+1}\left(O^{\left(i.l\right)}\right)$	Sequential Gauss–Newton $\left[\begin{array}{c} O^{\left(i,l+1\right)}\\ {\bar{O}}^{\left(i,l+1\right)} \end{array}\right]=\left[\begin{array}{c} O^{\left(i,l\right)}\\ {\bar{O}}^{\left(i,l\right)} \end{array}\right]-\left[\begin{array}{cc} Q_{l}^{*}diag\left(\frac{P}{{\left|P\right|}_{\max}}\right)Q_{l} & 0\\ 0 & Q_{l}^{T}diag\left(\frac{P}{{\left|P\right|}_{\max}}\right){\bar{Q}}_{l} \end{array}\right]{\left(H_{cc,l}^{A}\right)}^{-1}\left[\begin{array}{c} \nabla_{O}f_{A,l+1}\left(O^{\left(i,l\right)}\right)\\ \nabla_{\bar{O}}f_{A,l+1}\left(O^{\left(i,l\right)}\right) \end{array}\right]$
**Global**	Wirtinger flow $O^{\left(i+1\right)}=O^{\left(i\right)}-\alpha^{\left(i\right)}\nabla_{O}f_{A,l}\left(O^{\left(i\right)}\right)$	Global Gauss–Newton $\left[\begin{array}{c} O^{\left(i+1\right)}\\ {\bar{O}}^{\left(i+1\right)} \end{array}\right]=\left[\begin{array}{c} O^{\left(i\right)}\\ {\bar{O}}^{\left(i\right)} \end{array}\right]-\alpha^{\left(i\right)}{\left(H_{cc,l}^{A}\right)}^{-1}\left[\begin{array}{c} \nabla_{O}f_{A,l+1}\left(O^{\left(i\right)}\right)\\ \nabla_{\bar{O}}f_{A,l+1}\left(O^{\left(i\right)}\right) \end{array}\right]$

**Table 2 cells-11-01512-t002:** Suggested parameter ranges for ADMM-FPM method.

**Parameter**	** *α* **	** *β* **	** *γ* **	** *η* **
**Range**	0.5~1	1000	0.1~0.5	1

**Table 3 cells-11-01512-t003:** Runtime performance comparison of algorithms.

Approach	No Noise		Gaussian Noise		Poisson Noise	
	Iterations	Time	Iterations	Time	Iterations	Time
Gauss–Newton	72	21.84	35	10.33	46	15.13
mPIE	17	8.80	6	2.73	5	2.28
ADMM	80	35.54	42	15.81	50	20.82

## Data Availability

Data will be made available by the corresponding author on reasonable request.

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
