# Peer review of "Fourier Ptychographic Microscopy via Alternating Direction Method of Multipliers"

_cells, 2022, doi:10.3390/cells11091512_

Round 1

Reviewer 1 Report

Review on Fourier ptychographic microscopy
via alternating direction method of multipliers

by Aiye Wang, Zhuoqun Zhang, Siqi Wang, Caiwen Ma and Baoli Yao

The authors proposed the alternating direction method of multipliers (ADMM) for the
Fourier ptychographic imaging, concerned with the amplitude-Guassian metric based
nonlinear optimization problems. There are two major issues to be adressed.

First, such algorithm is not new for general phase retrieval problem, since there exist
several related work based on ADMMs (the group of Stefano Marchesini: [56] for
nonblind recovery and Chang et al. 2019 SIIMS for probe retrieval). It seems that the
authors simply extended such algorithm to Fourier ptychography, and it is not clear
about the contribution of this manuscript. I recommend the authors to first clearly state
what have been done in the mentioned references (may review the overall algorithms of
the references ahead), and then classify what contributions have been made in this
manuscirpt.

Another issue is the claim of ADMM as “second order algorithm. The author only
adopted the conventional version of ADMM which only led to the first-order
convergence in general. If they claimed the second-order property, the authors should
at least conduct a simulation to show the convergence speed, which is exactly second-
order (or at least not linear). Morover, based on their derivation for the subproblem in
the supplementary 1, it only showed the solver for the subproblems can be deeply
connected with second-order algorithm, that does not imply the second-order property
of the overall iterative scheme. Please either discuss this more precisely of the overall
scheme or remove such claim.

Other minor issues can be found below:

1. Can you add R-factor as a criterion besides SSIM
?
2. What is the role of the second term of equation (6) and (14)? Can you set up a
comparative experiment to illustrate?

3. How does ADMM perform compared to mPIE and Guass-Newton under
different amounts of LED light?

4. Line 150, what's the meaning of "weak-phase"?

5. What makes Figure 3 (b4
)and Figure 4 (c)phase of ADMM having such a big
difference?

6. In Experiments, ADMM has obvious light and shade difference compared with
the other two algorithms. For objective comparison, can you adjust them to gray
approximation?

7. Figure 5 (d) shows Guass-Newton and ADMM's convergence speed is very
different (Poisson noise σ = 104 ). Figure 4 (b1)and (b2)shows their very close to
the convergence (Poisson noise σ = 103 ). Can you measure a new experiment at
the same noise level
?

Reviewer 2 Report

Authors demonstrate the use of alternating direction method of multipliers (ADMM) in the field of Fourier ptychography microscopy (FPM). The results presented in the manuscript looks good. However, I have few concerns before accepting the paper for publication:

  1. Firstly, the major concern I have is that the results gets better with ADMM only after a certain level of noise in comparison to the other two existing techniques. Also, even after a certain noise Gauss-Newton method still gives similar performance as the proposed method. The improvement is not significant enough.
  2. Since four variables are updated in the iteration process of the proposed method, it is expected to give much better performance than the other two which further creates doubt about the use of ADMM. In my opinion, the author should focus on further optimizing the parameter in the simulation to achieve better efficiency.
  3. There are several grammatical errors in the manuscript which needs to be improved extensively.
  4. Some abbreviation are missing, such as PIE, LR. These should be defined for better understanding to the readers.
